# On the Rate of Convergence of Kolmogorov-Arnold Network Regression Estimators

## Abstract

Kolmogorov–Arnold Networks (KANs) offer a structured and interpretable framework for multivariate function approximation by composing univariate transformations through additive or multiplicative aggregation. This paper establishes theoretical convergence guarantees for KANs when the univariate components are represented by B-splines. We prove that both additive and hybrid additive–multiplicative KANs attain the minimax-optimal convergence rate $O(n^{-2r/(2r+1)})$ for functions in Sobolev spaces of smoothness $r$, where $n$ denotes the sample size. We further derive guidelines for selecting the optimal number of knots in the B-splines. The theory is supported by simulation studies that confirm the predicted convergence rates. These results provide a theoretical foundation for using KANs in nonparametric regression and highlight their potential as a structured alternative to existing methods.

## 1 Introduction

The convergence rates of nonparametric regression using neural networks have been extensively studied, building upon a long line of statistical learning theory (Barron, 2002; 1994; Imaizumi & Fukumizu, 2019; Bauer & Kohler, 2019; Schmidt-Hieber, 2020). Early foundational results, such as those of Stone (1982), established the minimax-optimal convergence rate for general nonparametric regression, which remains a benchmark for evaluating modern learning architectures. Subsequent advances in approximation theory demonstrated that fully connected deep neural networks, when endowed with sufficient width or depth, are capable of achieving rates close to these minimax limits. For instance, Kohler & Langer (2021) proved that deep neural networks can attain a rate of order $O((\log n)^6 n^{-\frac{2r}{2r+K}})$, thereby confirming their statistical efficiency under suitable smoothness conditions. While these results establish that neural networks are competitive with minimax-optimal procedures, their highly entangled representations generally lack explicit structure, limiting interpretability and complicating theoretical analysis.

In contrast, spline-based approximation methods provide a structured and well-understood pathway to minimax-optimality. The convergence behavior of spline estimators, particularly those constructed with B-splines, has been rigorously analyzed in both global and local settings (Speckman, 1985; Nussbaum, 1985; He & Shi, 1994; Li et al., 1995). Under standard smoothness assumptions, spline estimators achieve the optimal rate $O(n^{-\frac{2r}{2r+1}})$, which matches the minimax benchmark $O(n^{-\frac{2r}{2r+d}})$ in one-dimensional settings. Beyond splines, sieve estimators form a general class of nonparametric methods that approximate infinite-dimensional function spaces by growing finite-dimensional bases. Chen (2007) provided a comprehensive analysis of sieve estimation—including splines, wavelets, and polynomial bases—showing that these procedures can achieve minimax-optimal convergence when the basis is chosen adaptively to the function class. Similarly, kernel regression methods, such as the Nadaraya–Watson estimator and local polynomial regression, have long been known to reach minimax-optimal rates under appropriate bandwidth selection (Györfi et al., 2002). Collectively, these results underscore the value of structured nonparametric estimators: not only do they achieve optimal convergence rates, but their theoretical analysis also directly links model structure to approximation properties.

Kolmogorov–Arnold Networks (KANs) (Liu et al., 2024b) arise at the intersection of these two lines of research: like neural networks, they employ layered compositions of functions, but they inherit

the structured basis-function philosophy of spline and sieve methods. Their conceptual foundation lies in the Kolmogorov–Arnold representation theorem (Kolmogorov, 1957; Arnol'd, 1957), which asserts that any multivariate continuous function can be represented as a finite sum of continuous univariate functions applied to linear combinations of the input variables. This theorem provides a compositional, low-dimensional lens on high-dimensional learning problems, where complexity is managed through a hierarchy of simple univariate elements (Schmidt-Hieber, 2021). In practice, KANs instantiate this representation by parameterizing univariate components with B-spline expansions, thereby combining the expressive power of neural architectures with the interpretability and approximation guarantees of spline-based methods. The developments extend beyond purely additive structures to include multiplicative and hybrid architectures, improving expressiveness while retaining interpretability (Liu et al., 2024a). From a convergence perspective, this structure suggests that KANs may inherit the minimax-optimal rates known for spline estimators.

Despite their empirical success and conceptual appeal (Vaca-Rubio et al., 2024; Bodner et al., 2024; Kiamari et al., 2024; Koenig et al., 2024; Xu et al., 2024), the theoretical underpinnings of KANs remain underexplored. Universal approximation capabilities have been noted informally, but formal convergence analyses in statistical estimation settings are scarce. Recent work has begun to formalize this intuition. Gao & Tan (2025) studied the optimization and generalization behavior of KANs trained with stochastic gradient descent, providing non-asymptotic rate estimates that highlight how the structured nature of KANs influences their convergence. Unlike classical spline or kernel estimators, whose rates are derived from explicit approximation spaces, KANs introduce an additional layer of complexity through stochastic training dynamics. This makes their convergence properties more closely aligned with neural networks, where optimization and statistical considerations are tightly coupled. At the same time, the B-spline parameterization of univariate components offers a level of control and interpretability absent in generic neural networks. As a result, KANs occupy a distinctive position between classical sieve estimators and modern deep networks: they retain the statistical efficiency and minimax-optimal potential of spline-based methods, while leveraging the compositional flexibility of neural architectures.

We study both additive and hybrid additive–multiplicative KANs, modeling target functions as compositions of univariate spline transformations. We characterize the representable function classes and show that spline-based KAN estimators attain minimax-optimal convergence rates under standard Sobolev smoothness assumptions.

## 2 NONPARAMETRIC REGRESSION AND B-SPLINES

In nonparametric regression (Härdle, 1990; Tsybakov, 2008), the goal is to estimate an unknown function $f : [0,1]^d \to \mathbb{R}$ from observations $\{(X_i, Y_i)\}_{i=1}^n$ generated according to

$$Y_i = f(X_i) + \varepsilon_i, \quad i = 1, \ldots, n, \tag{1}$$

where $X_i = (X_{i1}, \ldots, X_{id}) \in [0,1]^d$ are the inputs and $\varepsilon_i$ are i.i.d. noise terms with zero mean and finite variance (e.g., $\varepsilon_i \sim \mathcal{N}(0, \sigma^2)$). Here, the domain $[0,1]^d$ is a standard choice for theoretical analysis without loss of generality (Tsybakov, 2008), since any bounded input domain $\mathcal{X} \in \mathbb{R}^d$ can be mapped to $[0,1]^d$ via an affine transformation.

Unlike parametric regression, which assumes a predetermined functional form for $f$ (e.g., linear or polynomial), the nonparametric setting imposes minimal structural assumptions, allowing the estimator to adapt to potentially complex and highly nonlinear relationships in the data. The central challenge lies in balancing this flexibility with statistical efficiency, particularly in high dimensions.

A common approach to nonparametric estimation is to approximate $f$ using a finite set of basis functions. Among the many available bases, B-splines are especially popular due to their computational efficiency, local support, and strong approximation properties under smoothness assumptions. In the univariate case, B-splines provide piecewise polynomial representations that achieve minimax-optimal convergence rates for sufficiently smooth $f$, and their tensor-product extensions naturally generalize to multivariate settings.

## 2.1 B-SPLINES IN NONPARAMETRIC REGRESSION

A univariate B-spline of order $p + 1$ (corresponding to polynomial degree $p$) is defined as a linear combination of basis functions $\{B_{i,p}\}_{i=0}^{K+p}$ associated with a non-decreasing knot sequence $\{t_0, t_1, \ldots, t_K\}$. Each basis function $B_{i,p}$ is a piecewise polynomial with continuous differentiability up to order $p - 1$ (Schumaker, 2007). These basis functions possess local support, being nonzero only over an interval determined by $p + 1$ consecutive knots. This ensures that modifying one coefficient influences the spline only on a limited subinterval, thereby improving both interpretability and numerical stability.

The B-spline basis functions are constructed iteratively using the Cox–de Boor recursion formula:

$$B_{i,0}(x) = \begin{cases} 1, & t_i \leq x < t_{i+1}, \\ 0, & \text{otherwise,} \end{cases} \tag{2}$$

$$B_{i,p}(x) = \frac{x - t_i}{t_{i+p} - t_i} B_{i,p-1}(x) + \frac{t_{i+p+1} - x}{t_{i+p+1} - t_{i+1}} B_{i+1,p-1}(x), \quad p \geq 1, \tag{3}$$

where division by zero is interpreted as zero.

The basis functions form a partition of unity,

$$\sum_{i=0}^{K+p} B_{i,p}(x) = 1, \quad \forall x \in [0, 1]. \tag{4}$$

Hence, any spline function $f$ of order $p + 1$ can be expressed as

$$f(x) = \sum_{i=0}^{K+p} c_i B_{i,p}(x), \tag{5}$$

where the coefficients $\{c_i\}$ are estimated from data, typically via least squares or penalized regression.

B-splines achieve minimax-optimal convergence rates under standard Sobolev smoothness conditions and extend naturally to higher dimensions via tensor-product constructions (Unser et al., 2002). In the context of Kolmogorov–Arnold Networks, these properties make B-splines a natural choice for modeling the learnable univariate transformations.

## 2.2 MINIMAX OPTIMAL CONVERGENCE RATE

In nonparametric regression, the performance of an estimator $\hat{f}_n$ is often characterized by how quickly it converges to the true regression function $f$ as the sample size $n$ increases. The *minimax optimal convergence rate* describes the best achievable rate, under worst-case conditions, across all possible estimators for a given function class.

Let $\mathcal{F}$ be a class of functions (e.g., a Sobolev space $W^r([0, 1]^d)$), and let $\hat{f}_n$ be an estimator of $f \in \mathcal{F}$. The minimax risk is defined as (Györfi et al., 2002)

$$R_{\text{minimax}} = \inf_{\hat{f}_n} \sup_{f \in \mathcal{F}} \mathbb{E}\left[\|\hat{f}_n - f\|_{L^2([0,1]^d)}^2\right], \tag{6}$$

where $\|g\|_{L^2([0,1]^d)}^2 = \int_{[0,1]^d} |g(x)|^2 \, dx$ denotes the squared $L^2$-norm over the domain $[0, 1]^d$.

The minimax optimal rate $\mathcal{R}_{\text{minimax}}(n)$ describes how quickly the risk decays as the sample size $n$ increases. For functions in the Sobolev space $W^r([0, 1]^d)$ of smoothness $r > 0$, it is well known (Stone, 1982) that

$$\inf_{\hat{f}_n} \sup_{f \in W^r([0,1]^d)} \mathbb{E}\left[\|\hat{f}_n - f\|_{L^2}^2\right] = O\left(n^{-\frac{2r}{2r+d}}\right). \tag{7}$$

Here, $n$ denotes the sample size, $r$ the smoothness index of $f$, and $d$ the input dimension. This exponent reflects the trade-off between smoothness $r$ and dimensionality $d$: smoother functions are easier to estimate, while higher dimensions slow convergence.

This expression makes explicit the *curse of dimensionality*: the convergence rate deteriorates rapidly with increasing $d$, even when $f$ is very smooth. In particular, when $d$ is large, achieving a small estimation error requires exponentially more samples, motivating the design of estimators that can exploit low-dimensional structures in high-dimensional data.

### 2.3 MINIMAX OPTIMALITY OF B-SPLINES IN NONPARAMETRIC REGRESSION

When the target function is assumed to belong to a Sobolev space $W^r([0,1])$ of smoothness order $r > 0$, the minimax optimal convergence rate in this one-dimensional setting is known to be $O\left(n^{-\frac{2r}{2r+1}}\right)$, which is attained by a broad class of nonparametric estimators, including kernel methods, local polynomials, and spline-based approaches.

B-spline estimators are a particularly appealing choice for achieving this optimal rate, owing to their locality, numerical stability, and flexibility in representing smooth functions. Classical results Speckman (1985); He & Shi (1994); Li et al. (1995) have established that, when the number of spline basis functions is chosen appropriately (balancing approximation and estimation errors), an estimator

$$\hat{f}_n(x) = \sum_{i=0}^{K+p} \hat{c}_i B_{i,p}(x), \tag{8}$$

where $\{B_{i,p}\}_{i=0}^{K+p}$ are B-spline basis functions of degree $p \geq r$, achieves the minimax rate

$$\sup_{f \in W^r([0,1])} \mathbb{E}\big[\|\hat{f}_n - f\|_2^2\big] = O\left(n^{-\frac{2r}{2r+1}}\right). \tag{9}$$

This result is minimax optimal in the sense that no estimator can uniformly outperform this rate over the entire Sobolev ball $\{f \in W^r : \|f\|_{W^r} \leq C\}$. The optimality holds regardless of the specific B-spline degree, provided it is sufficiently high to capture the prescribed smoothness, and hinges on balancing the spline knot density with the available sample size.

Table 1: Comparison of convergence rates for different nonparametric estimators. Here $r$ denotes smoothness of the target function, $d$ the input dimension, and $n$ the sample size.

| Method | Convergence Rate | References |
|---|---|---|
| B-splines | $O\left(n^{-\frac{2r}{2r+1}}\right)$ | Speckman (1985) |
| Sieve estimators | $O\left(n^{-\frac{2r}{2r+d}}\right)$ | Chen (2007) |
| Kernel regression | $O\left(n^{-\frac{2r}{2r+d}}\right)$ | Györfi et al. (2002) |
| Deep neural networks (fully connected) | $O\left((\log n)^6 n^{-\frac{2r}{2r+K}}\right)$ | Kohler & Langer (2021) |
| Kolmogorov–Arnold Networks (KANs) | Under investigation | Liu et al. (2024b) |

The significance of this property extends to structured neural architectures such as Kolmogorov–Arnold Networks (KANs), where B-spline parameterizations can serve as the univariate building blocks. In such cases, the known optimality of B-splines in one-dimensional regression informs the achievable convergence rates for the multivariate composite model. In the next section, we formalize this connection and prove that KAN estimators inherit the minimax optimal convergence rate $O\left(n^{-\frac{2r}{2r+1}}\right)$ when the target function lies in a Sobolev space with smoothness $r$. A summary of representative convergence results across classical and modern nonparametric estimators is provided in Table 1.

## 3 MODEL SETUP

Let $x = (x_1, \ldots, x_d) \in [0,1]^d$, and consider the problem of approximating a multivariate function $f : [0,1]^d \to \mathbb{R}$. A Kolmogorov–Arnold Network (KAN) models $f$ as a finite sum of univariate

nonlinearities composed with structured aggregations of input variables:

$$f(x) = \sum_{q=1}^{Q} g_q\left(T_q(x)\right), \tag{10}$$

where each $g_q : \mathbb{R} \to \mathbb{R}$ is a univariate activation function, and each $T_q(x)$ is a structured transformation of the inputs. Specifically, we consider two forms of $T_q(x)$:

$$T_q(x) = \begin{cases} \sum_{j=1}^{d} \psi_{qj}(x_j), & \text{(additive node)} \\ \prod_{j=1}^{d} \psi_{qj}(x_j), & \text{(multiplicative node)} \end{cases} \tag{11}$$

where the functions $\psi_{qj} : [0,1] \to \mathbb{R}$ are univariate spline transformations applied to individual input dimensions. This architecture captures both additive and multiplicative interactions among input features, enabling greater expressiveness while maintaining interpretability.

Each univariate function $\psi_{qj}$ and $g_q$ is parameterized by a finite B-spline basis. Let $W^r([0,1])$ denote the Sobolev space of univariate functions with $r$ square-integrable derivatives. We assume that each $\psi_{qj}$ and $g_q$ belongs to $W^r([0,1])$ for some fixed smoothness index $r > \frac{1}{2}$, which ensures the Sobolev embedding $W^r([0,1]) \subset C([0,1])$, so that all univariate components are continuous and the compositions are well defined.

We define the following function classes associated with KANs:

- **Additive KAN class** $\mathcal{F}_{\text{add}}$: functions of the form

$$f(x) = \sum_{q=1}^{Q} g_q\left(\sum_{j=1}^{d} \psi_{qj}(x_j)\right), \quad \text{with } \psi_{qj}, g_q \in W^r([0,1]). \tag{12}$$

- **Hybrid KAN class** $\mathcal{F}_{\text{hyb}}$: functions where each node may use either an additive or multiplicative composition,

$$f(x) = \sum_{q=1}^{Q} g_q\left(T_q(x)\right), \quad \text{with } T_q(x) \in \left\{\sum_{j=1}^{d} \psi_{qj}(x_j),\ \prod_{j=1}^{d} \psi_{qj}(x_j)\right\}. \tag{13}$$

Throughout this paper, we assume that all univariate functions $\psi_{qj}$ and $g_q$ belong to $W^r([0,1])$ with $r \geq 1$, are uniformly bounded in sup norm by a constant $M$, and that each $g_q$ is Lipschitz on its range with Lipschitz constant bounded by $L$. These regularity conditions are standard in nonparametric regression and ensure statistical well-posedness of the KAN estimator.

The KAN model is learned from data $\{(X_i, Y_i)\}_{i=1}^{n}$, where $Y_i = f(X_i) + \varepsilon_i$, and $\varepsilon_i$ are noises with zero mean and finite variance. For a given sample size $n$, the collection of all such spline-parameterized KANs spans a finite-dimensional function class. We define the corresponding estimator $\hat{f}_n$ as the empirical risk minimizer of the squared loss over this class, i.e., a least-squares fit of the KAN model to the data. We refer to this empirical risk minimizer as the **spline-based KAN sieve estimator**.

**Sieve formulation of the estimator.** For theoretical analysis, we view the spline–parameterized KAN as a sieve estimator. For each sample size $n$, let $\mathcal{F}_n$ denote the finite–dimensional KAN spline space obtained by parameterizing every univariate component $g_q$ and $\psi_{qj}$ with B-spline bases of order $m \geq r$ and $k_n$ interior knots. Thus $\dim(\mathcal{F}_n) \asymp k_n$ under fixed $(Q, d)$. The estimator introduced above can therefore be written as the empirical risk minimizer

$$\hat{f}_n \in \arg\min_{f \in \mathcal{F}_n} \frac{1}{n} \sum_{i=1}^{n} (Y_i - f(X_i))^2.$$

The sequence $\{\mathcal{F}_n\}_{n \geq 1}$ forms a growing family of approximation spaces whose union is dense in the KAN function class $\mathcal{F}_r^{\text{KAN}}$, so $\hat{f}_n$ is a sieve least–squares estimator in the sense of classical nonparametric sieve theory. Importantly, the statistical results in Section 4 concern this population estimator, independent of the numerical optimization procedure used to approximately solve the least–squares problem in practice.

This section sets up the framework for our main theoretical results, which establish optimal convergence rates and structural guarantees for KAN estimators under these modeling assumptions.

## 4 MAIN RESULTS

We now present the theoretical convergence properties of Kolmogorov–Arnold Networks (KANs) under the regression setting introduced in Section 3. Our goal is to quantify the estimation error of the spline-based KAN estimator $\hat{f}_n$, defined as the empirical risk minimizer of the squared loss over the spline-parameterized KAN sieve function class introduced in Section 3, where

$$Y_i = f(X_i) + \varepsilon_i, \quad \mathbb{E}[\varepsilon] = 0, \quad \mathbb{E}[\varepsilon^2] = \sigma^2 < \infty. \tag{14}$$

We consider two architectures: additive KANs, where each node aggregates input features additively, and hybrid additive–multiplicative KANs, where node-level interactions are either additive or multiplicative. In both settings, the univariate component functions $\psi_{qj}$ and $g_q$ are assumed to belong to the Sobolev space $W^r([0,1])$ with smoothness $r \geq 1$, are uniformly bounded, and the outer functions $g_q$ are Lipschitz.

The following results establish convergence rates for $\hat{f}_n$ in the $L^2$ risk. We show that additive KANs attain the classical minimax rate for univariate nonparametric regression, and that hybrid KANs retain this rate up to a constant overhead.

**Theorem 1** (Convergence rate of additive KAN with spline sieve). *Let $\hat{f}_n$ be the spline–based KAN sieve estimator defined in Section 3, and suppose that the true regression function $f_0$ is of the additive KAN form*

$$f_0(x) = \sum_{q=1}^{Q} g_q \left( \sum_{j=1}^{d} \psi_{qj}(x_j) \right)$$

*with $g_q, \psi_{qj} \in W^r([0,1])$ and $r > 1/2$, then*

$$\mathbb{E}\left[ \|\hat{f}_n - f_0\|_{L^2([0,1]^d)}^2 \right] = O\left( n^{-\frac{2r}{2r+1}} \right). \tag{15}$$

*Remark* 1. This result establishes that additive KANs achieve the minimax optimal convergence rate for functions in the additive Sobolev class with smoothness index $r$. The rate $n^{-\frac{2r}{2r+1}}$ corresponds to the optimal rate for estimating univariate functions in $W^r([0,1])$. Notably, this convergence rate does not depend on the ambient dimensionality $d$, thereby avoiding the curse of dimensionality. This property arises from the compositional structure of KANs, which represent multivariate functions as sums of univariate functions constructed from B-splines.

Following the interpretation above, it is natural to ask whether the decomposition of $f_0(x) = \sum_{q=1}^{Q} g_q \left( \sum_{j=1}^{d} \psi_{qj}(x_j) \right)$ into univariate components is uniquely determined. Although identifiability is not required for the statistical convergence in Theorem 1, it plays an important role in the interpretability of the learned univariate components. The next proposition resolves this question.

**Proposition 1** (Identifiability up to constant shifts). *Let*

$$f(x) = \sum_{q=1}^{Q} g_q \left( \sum_{j=1}^{d} \psi_{qj}(x_j) \right) \tag{16}$$

*be a KAN function defined on $[0,1]^d$, where $g_q$ and $\psi_{qj}$ are univariate measurable functions. Suppose further that for each $q$ we apply the normalizations*

$$\int_0^1 \psi_{qj}(t)\,dt = 0, \qquad j = 1, \dots, d, \tag{17}$$

*and*

$$\int_0^1 g_q(u)\,d\mu_q(u) = 0, \tag{18}$$

*where $\mu_q$ is the distribution of $\sum_{j=1}^{d} \psi_{qj}(X_j)$. Then the representation of $f$ in terms of $\{g_q, \psi_{qj}\}$ is identifiable up to permutation of the $Q$ units. Conversely, without the centering constraints*

equation 17–equation 18, the representation is not identifiable: for any constants $c_{qj}$ satisfying $\sum_{j=1}^{d} c_{qj} = 0$,

$$\psi_{qj}(x_j) \mapsto \psi_{qj}(x_j) + c_{qj}, \qquad g_q(u) \mapsto g_q(u - \sum_{j=1}^{d} c_{qj}) \tag{19}$$

leaves $f$ unchanged.

*Remark* 2. Proposition 1 shows that KAN representations are generally non-identifiable unless constant-shift ambiguities are removed. This has no impact on the convergence results in Theorem 1, because statistical estimation is performed over the full sieve class $\mathcal{F}_n$, and the risk $\|f - f_0\|_{L^2}$ is unaffected by reparameterizations of the internal functions $\psi_{qj}$ and $g_q$. In practice, optimization algorithms automatically pick one admissible parameterization among many equivalent ones, and identifiability matters only when the goal is to interpret the learned univariate components. Imposing centering constraints as in Proposition 1 guarantees unique decomposition and therefore improves interpretability, but is not required for the statistical consistency or the convergence rate of $\hat{f}_n$.

We now extend the analysis to hybrid KANs, which permit more expressive architectures by allowing either additive or multiplicative compositions within each node.

**Theorem 2** (Convergence rate of hybrid KAN with spline sieve). *Let $\hat{f}_n$ be the spline–based KAN sieve estimator defined in Section 3, and assume that the true regression function has the multiplicative KAN form*

$$f_0(x) = \sum_{q=1}^{Q} g_q \left( \prod_{j=1}^{d} \psi_{qj}(x_j) \right)$$

*with $g_q, \psi_{qj} \in W^r([0,1])$ and $r > 1/2$, then*

$$\mathbb{E}\left[ \|\hat{f}_n - f_0\|_{L^2([0,1]^d)}^2 \right] = O\left( n^{-\frac{2r}{2r+1}} \right). \tag{20}$$

*Remark* 3. The hybrid architecture allows for both additive and multiplicative compositions of univariate functions. The multiplicative terms induce a constant overhead of $M^{2(d-1)}$, but the convergence rate in $n$ remains minimax optimal. This overhead arises because estimating multiplicative interactions requires products of univariate spline bases. While this increases the constant factor, the dependence on $n$ is unaffected. For moderate dimensions and bounded smooth components, hybrid KANs remain statistically efficient.

Theorems 1 and 2 provide a theoretical explanation for recent empirical findings (Wang et al., 2025; SS et al., 2024; Aghaei, 2024), where KANs achieve near-minimal losses across diverse practical tasks. The following corollary formalizes the minimax optimality of KAN estimators over a broad class of structured functions, encompassing both additive and hybrid compositions.

**Corollary 1** (Minimax optimality of KAN estimators). *Fix $r > 1/2$ and let $\mathcal{F}_r^{\text{KAN}}$ denote the class of continuous functions $f : [0,1]^d \to \mathbb{R}$ that admit a KAN representation with univariate Sobolev smoothness $r$, i.e.,*

$$\mathcal{F}_r^{\text{KAN}} = \left\{ f : f(x) = \sum_{q=1}^{Q} g_q\big(T_q(x)\big), \; g_q, \psi_{qj} \in W^r([0,1]), \; T_q(x) \in \{\sum_{j=1}^{d} \psi_{qj}(x_j), \prod_{j=1}^{d} \psi_{qj}(x_j)\} \right\}, \tag{21}$$

*where $Q$ and $d$ are fixed and the sums/products are taken over univariate components $\psi_{qj}$. Consider the minimax risk*

$$R_n(\mathcal{F}_r^{\text{KAN}}) = \inf_{\hat{f}_n} \sup_{f_0 \in \mathcal{F}_r^{\text{KAN}}} \mathbb{E}_{f_0}\left[ \|\hat{f}_n - f_0\|_{L^2([0,1]^d)}^2 \right], \tag{22}$$

*where the infimum is over all estimators $\hat{f}_n$ based on $n$ i.i.d. observations from the regression model $Y = f_0(X) + \varepsilon$, $\mathbb{E}[\varepsilon \mid X] = 0$, $\mathbb{E}[\varepsilon^2] < \infty$. Then*

$$R_n(\mathcal{F}_r^{\text{KAN}}) \asymp n^{-\frac{2r}{2r+1}}, \tag{23}$$

*i.e., there exist constants $0 < c \leq C < \infty$ independent of $n$ such that*

$$c \, n^{-\frac{2r}{2r+1}} \;\leq\; R_n(\mathcal{F}_r^{\text{KAN}}) \;\leq\; C \, n^{-\frac{2r}{2r+1}}. \tag{24}$$

*In particular, the spline-based KAN sieve estimators of Theorems 1 and 2 are minimax optimal over $\mathcal{F}_r^{\text{KAN}}$.*

Finally, we provide a guideline for choosing the number of knots in the spline basis for optimal performance.

**Corollary 2** (Optimal knot number for spline-based KAN sieves). *Let $r > 1/2$, and suppose each univariate component $g_q$ and $\psi_{qj}$ in the KAN architecture is represented by a B-spline basis of order $m \geq r$ with $k_n$ interior knots, as in the sieve spaces $\mathcal{F}_n$ used in Theorems 1 and 2. Then the choice*

$$k_n \asymp n^{1/(2r+1)} \tag{25}$$

*balances the approximation error and estimation error in the KAN sieve estimator and yields the minimax-optimal convergence rate*

$$\mathbb{E}\big[\|\hat{f}_n - f_0\|_{L^2([0,1]^d)}^2\big] = O\Big(n^{-\frac{2r}{2r+1}}\Big). \tag{26}$$

*In particular, for each univariate spline unit approximating a component in $W^r([0,1])$, the resulting squared $L^2$ error satisfies*

$$\mathbb{E}\big[\|\hat{\psi}_{qj} - \psi_{qj}\|_{L^2([0,1])}^2\big] = O\Big(n^{-\frac{2r}{2r+1}}\Big) \tag{27}$$

*under the global knot choice equation 68.*

*Remark* 4. This result follows from classical nonparametric regression theory and provides a principled guideline for selecting the spline resolution in KANs based on the sample size and assumed smoothness of the target function.

These results confirm that spline-based KAN estimators achieve minimax-optimal convergence under standard smoothness assumptions. While additive KANs offer statistical efficiency and simplicity, hybrid architectures expand expressiveness with only a modest complexity overhead. These theoretical guarantees provide a strong foundation for using KANs in structured, interpretable learning tasks.

## 5 SIMULATION STUDY

To verify the theoretical results and demonstrate the practical performance of Kolmogorov–Arnold Networks (KANs), we conduct simulation experiments on synthetic datasets generated from functions with known smoothness. It is noted that this paper focuses on theoretical convergence rather than empirical benchmarking, and the experiments are intentionally designed to validate that the empirical convergence rate matches the theoretical exponent. In particular, we compare additive KANs and hybrid additive–multiplicative KANs against standard multilayer perceptrons (MLPs). Including the MLP baseline serves to verify the theoretical contrast between KAN and conventional neural networks in terms of convergence behavior.

### 5.1 EXPERIMENTAL SETUP

Synthetic regression data are generated from functions defined on $[0,1]^d$. We consider two representative targets.

The first target is a piecewise polynomial function with $d = 5$ and exact Sobolev smoothness $r$, given by

$$f(x) = \sin\Big(\pi \sum_{j=1}^{d} \psi(x_j)\Big), \tag{28}$$

where $\psi : [0,1] \to \mathbb{R}$ is defined as

$$\psi(t) = \begin{cases} t^{r+1}, & 0 \leq t < \frac{1}{2}, \\ (1-t)^{r+1}, & \frac{1}{2} \leq t \leq 1. \end{cases}$$

The second target is a periodic function with $d = 1$ and exact Sobolev smoothness $r$, constructed as a Fourier series with coefficients

$$a_m = \frac{1}{m^{r+1/2} \log(m+1)}, \qquad m \geq 1,$$

so that

$$f(x) \; = \; \sum_{m=1}^{\infty} a_m \sin(2\pi mx).$$

To see that $f$ has Sobolev smoothness exactly $r$, recall that for a Fourier series with coefficients $a_m$, membership $f \in W^t([0,1])$ is equivalent (up to constants) to

$$\sum_{m=1}^{\infty} (1+m^2)^t |a_m|^2 < \infty.$$

Substituting $a_m = 1/(m^{r+1/2} \log(m+1))$, for large $m$ the summand behaves like

$$(1+m^2)^t |a_m|^2 \sim m^{2t} \cdot m^{-2r-1} \cdot \frac{1}{\log^2 m} = \frac{m^{2(t-r)-1}}{\log^2 m}.$$

If $t = r$, the summand is $\sim 1/(m \log^2 m)$, and the series converges by the standard Cauchy condensation test (the condensed summand is $\sim 1/m^2$). Hence $f \in W^r([0,1])$.

If $t > r$, the exponent $2(t-r)-1 > -1$, so the summand behaves like $m^\alpha$ with $\alpha \geq -1 + \delta$ for some $\delta > 0$, and the series diverges. Hence $f \notin W^t([0,1])$ for any $t > r$.

Therefore, $f$ belongs to $W^r([0,1])$ but not to any $W^{r+\varepsilon}([0,1])$, i.e., its Sobolev smoothness is exactly $r$. This construction provides a controlled setting for evaluating estimators precisely at the smoothness condition.

In the experiments, we use $r = 2$, which for the polynomial case yields a piecewise cubic function with continuous first and second derivatives. Training inputs are sampled uniformly as $X_i \sim \mathsf{Unif}([0,1]^d)$, and responses are observed as

$$Y_i \; = \; f(X_i) + \varepsilon_i, \qquad \varepsilon_i \sim \mathcal{N}(0, \sigma^2), \quad \sigma = 0.05. \tag{29}$$

Sample sizes are chosen as $n \in \{100, 200, 400, 800, 1600, 3200, 6400, 12800\}$. For each $n$, the number of B-spline grid intervals is set according to the theoretical guideline $k \asymp n^{1/(2r+1)}$. Both additive and hybrid KANs are constructed using cubic ($p = 3$) B-splines, and their parameters are fitted by minimizing the empirical squared loss. All models are optimized using the LBFGS algorithm (Liu & Nocedal, 1989), which in this setting serves as a numerical solver for the least-squares problem defining the spline-based KAN sieve estimator.

## 5.2 RESULTS

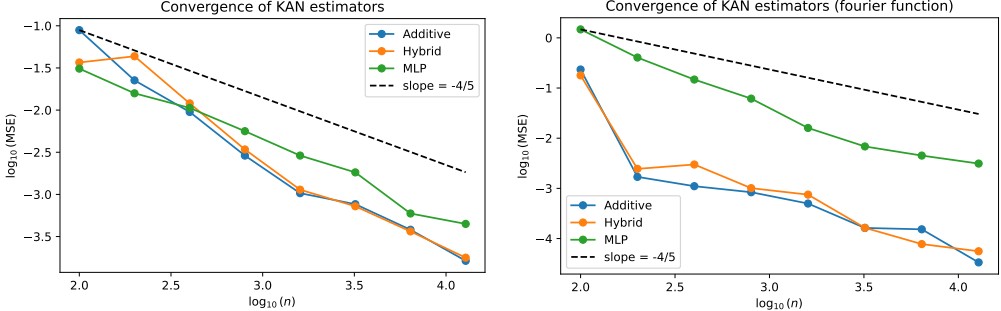

Figure 1: Convergence rates of additive KAN, hybrid KAN, and MLP estimators on two synthetic targets. The dashed line indicates the theoretical slope $-4/5$. Left: piecewise polynomial function with exact Sobolev smoothness $r = 2$. Right: periodic function with exact Sobolev smoothness $r = 2$, constructed from a Fourier series. Mean squared error on a test set is plotted against sample size $n$ on a log–log scale.

Figure 1 compares the convergence behavior of the three estimators across the two function classes. In both cases, additive and hybrid KANs achieve slopes that closely follow, and in fact are steeper

than, the theoretical rate. This is consistent with the theory, since the minimax-optimal rate represents a worst-case upper bound. In favorable instances, such as the structured piecewise polynomial and Fourier series functions considered here, empirical performance can exceed this baseline.

The two KAN variants exhibit nearly indistinguishable performance as sample sizes increase, indicating that the inclusion of multiplicative interactions does not substantially alter convergence in these settings. By contrast, the MLP baseline converges more slowly and requires much larger sample sizes to approach the accuracy attained by KANs.

These results validate the theoretical analysis and confirm that spline-based KANs achieve both statistical optimality and practical efficiency. The convergence pattern highlights the advantage of incorporating structural priors through B-spline representations, which enables KANs to learn efficiently even with moderate data, in contrast to the slower learning dynamics observed in black-box neural networks such as MLPs.

# 6  CONCLUSION

This paper has presented a theoretical analysis of Kolmogorov–Arnold Networks (KANs) with univariate spline units for nonlinear regression. We established that both additive and hybrid additive–multiplicative KAN architectures achieve minimax-optimal convergence rates under standard smoothness assumptions.

The theoretical guarantees are corroborated by simulation experiments, which show that both additive and hybrid KANs empirically attain the optimal convergence rate, and perform comparably across sample sizes. In contrast, standard neural networks exhibit slower convergence and require larger data to achieve similar accuracy.

The theoretical results established in this work concern the spline-parameterized KAN estimator defined through empirical risk minimization on a growing sieve space. They do not rely on the dynamics of stochastic or gradient-based training and remain valid as long as numerical optimization returns (or sufficiently approximates) the empirical minimizer. Extending the analysis to characterize convergence properties under specific optimization algorithms, such as SGD or alternating backfitting, represents an important direction for future work.

Future work will also explore scalable algorithmic implementations of KANs for real-world datasets and integration with deep learning architectures for structured end-to-end system dynamics learning and symbolic equation discovery.

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

## A APPENDIX: PROOFS OF THEOREMS AND COROLLARIES

### A.1 PROOF OF THEOREM 1: CONVERGENCE RATE OF ADDITIVE KAN

**Theorem 1** (Convergence rate of additive KAN with spline sieve). *Let $\hat{f}_n$ be the spline–based KAN sieve estimator defined in Section 3, and suppose that the true regression function $f_0$ is of the additive KAN form*

$$f_0(x) = \sum_{q=1}^{Q} g_q\left(\sum_{j=1}^{d} \psi_{qj}(x_j)\right)$$

*with $g_q, \psi_{qj} \in W^r([0,1])$ and $r > 1/2$, then*

$$\mathbb{E}\left[\|\hat{f}_n - f_0\|_{L^2([0,1]^d)}^2\right] = O\left(n^{-\frac{2r}{2r+1}}\right). \tag{30}$$

*Proof.* Let $\|\cdot\|$ denote the $L^2([0,1]^d)$ norm. For each sieve space $\mathcal{F}_n$, define

$$f_n^* = \arg\min_{f \in \mathcal{F}_n} \|f - f_0\|. \tag{31}$$

Using the standard decomposition,

$$\|\hat{f}_n - f_0\| \le \|\hat{f}_n - f_n^*\| + \|f_n^* - f_0\|, \tag{32}$$

we handle the two terms separately.

**Step 1: Approximation error.** Since $g_q, \psi_{qj} \in W^r([0,1])$ and $m \ge r$, there exist spline approximations such that (standard spline approximation theory)

$$\|g_q - g_{q,k_n}\|_{L^2} = O(k_n^{-r}), \qquad \|\psi_{qj} - \psi_{qj,k_n}\|_{L^2} = O(k_n^{-r}). \tag{33}$$

For each $q$, define the spline-approximated component

$$f_{0,q,k_n}(x) = g_{q,k_n}\left(\sum_{j=1}^{d} \psi_{qj,k_n}(x_j)\right). \tag{34}$$

Using the Lipschitz continuity of $g_q$ on the compact range of $\sum_j \psi_{qj}$,

$$\|f_{0,q,k_n} - f_{0,q}\| \le L_q \left\|\sum_{j=1}^{d}(\psi_{qj,k_n} - \psi_{qj})\right\| \le C \sum_{j=1}^{d} \|\psi_{qj,k_n} - \psi_{qj}\| \tag{35}$$

$$= O(k_n^{-r}). \tag{36}$$

Summing over $q = 1, \ldots, Q$ yields

$$\|f_n^* - f_0\|^2 = O(k_n^{-2r}). \tag{37}$$

**Step 2: Estimation error.** Let $p_n$ denote the total number of spline coefficients across all $g_{q,k_n}$ and $\psi_{qj,k_n}$. Since $Q$ and $d$ are fixed,

$$p_n \asymp k_n. \tag{38}$$

The class $\mathcal{F}_n$ is quadratic-loss parametric with dimension $p_n$, so its metric entropy satisfies

$$\log N(\epsilon, \mathcal{F}_n, \|\cdot\|_\infty) \lesssim p_n \log(1/\epsilon). \tag{39}$$

By a standard empirical-process argument for sieve least squares (Chen 2007; Györfi et al. 2002; Newey and McFadden 1994), this implies

$$\mathbb{E}[\|\hat{f}_n - f_n^*\|^2] = O\left(\frac{p_n}{n}\right). \tag{40}$$

Using equation 38, this becomes

$$\mathbb{E}[\|\hat{f}_n - f_n^*\|^2] = O\left(\frac{k_n}{n}\right). \tag{41}$$

**Step 3: Rate balancing.** Combining equation 32, equation 37, and equation 40,

$$\mathbb{E}\left[\|\hat{f}_n - f_0\|^2\right] \leq C\left(k_n^{-2r} + \frac{k_n}{n}\right). \tag{42}$$

The optimal balance is $k_n \asymp n^{1/(2r+1)}$, resulting in

$$\mathbb{E}\left[\|\hat{f}_n - f_0\|^2\right] = O\left(n^{-\frac{2r}{2r+1}}\right). \tag{43}$$

This proves the theorem. □

### A.2 PROOF OF THEOREM 2: CONVERGENCE RATE OF HYBRID ADDITIVE–MULTIPLICATIVE KAN

**Theorem 2** (Convergence rate of hybrid KAN with spline sieve). *Let $\hat{f}_n$ be the spline–based KAN sieve estimator defined in Section 3, and assume that the true regression function has the multiplicative KAN form*

$$f_0(x) = \sum_{q=1}^{Q} g_q\left(\prod_{j=1}^{d} \psi_{qj}(x_j)\right)$$

*with $g_q, \psi_{qj} \in W^r([0,1])$ and $r > 1/2$, then*

$$\mathbb{E}\left[\|\hat{f}_n - f_0\|^2_{L^2([0,1]^d)}\right] = O\left(n^{-\frac{2r}{2r+1}}\right). \tag{44}$$

*Proof.* Let

$$f_{0,q}(x) = g_q\left(\prod_{j=1}^{d} \psi_{qj}(x_j)\right). \tag{45}$$

Define $f_n^*$ as in the previous proof. We again apply

$$\|\hat{f}_n - f_0\| \leq \|\hat{f}_n - f_n^*\| + \|f_n^* - f_0\|. \tag{46}$$

**Step 1: Approximation error.** Using the same spline approximation bounds as in Theorem 1,

$$\|\psi_{qj} - \psi_{qj,k_n}\| = O(k_n^{-r}), \qquad \|g_q - g_{q,k_n}\| = O(k_n^{-r}). \tag{47}$$

Because all functions are bounded on $[0,1]$, denote $M = \max_{q,j} \|\psi_{qj}\|_\infty$. Then

$$\left\|\prod_{j=1}^{d} \psi_{qj} - \prod_{j=1}^{d} \psi_{qj,k_n}\right\| \leq C \sum_{j=1}^{d} \|\psi_{qj} - \psi_{qj,k_n}\| \tag{48}$$

$$= O(k_n^{-r}). \tag{49}$$

Applying the Lipschitz continuity of $g_q$ on the compact range of the products,

$$\|g_q(\prod_j \psi_{qj}) - g_{q,k_n}(\prod_j \psi_{qj,k_n})\| = O(k_n^{-r}), \tag{50}$$

and summation over $q$ yields

$$\|f_n^* - f_0\|^2 = O(k_n^{-2r}). \tag{51}$$

**Step 2: Estimation error.** The number of spline coefficients is unchanged: $p_n \asymp k_n$. Thus the same sieve LS bound applies:

$$\mathbb{E}[\|\hat{f}_n - f_n^*\|^2] = O\left(\frac{p_n}{n}\right) = O\left(\frac{k_n}{n}\right). \tag{52}$$

**Step 3: Rate balancing.** Finally,

$$\mathbb{E}\left[\|\hat{f}_n - f_0\|^2\right] \leq C\left(k_n^{-2r} + \frac{k_n}{n}\right). \tag{53}$$

Choosing $k_n \asymp n^{1/(2r+1)}$ balances the two terms and yields

$$\mathbb{E}\left[\|\hat{f}_n - f_0\|^2\right] = O\left(n^{-\frac{2r}{2r+1}}\right). \tag{54}$$

□

## A.3 Proof of Corollary 1

**Corollary 1** (Minimax optimality of KAN estimators). *Fix $r > 1/2$ and let $\mathcal{F}_r^{\text{KAN}}$ denote the class of continuous functions $f : [0,1]^d \to \mathbb{R}$ that admit a KAN representation with univariate Sobolev smoothness $r$, i.e.,*

$$\mathcal{F}_r^{\text{KAN}} = \Big\{ f : f(x) = \sum_{q=1}^{Q} g_q\Big(T_q(x)\Big), \; g_q, \psi_{qj} \in W^r([0,1]), \; T_q(x) \in \{\textstyle\sum_{j=1}^d \psi_{qj}(x_j), \prod_{j=1}^d \psi_{qj}(x_j)\} \Big\}, \tag{55}$$

*where $Q$ and $d$ are fixed and the sums/products are taken over univariate components $\psi_{qj}$. Consider the minimax risk*

$$R_n(\mathcal{F}_r^{\text{KAN}}) = \inf_{\hat{f}_n} \sup_{f_0 \in \mathcal{F}_r^{\text{KAN}}} \mathbb{E}_{f_0}\Big[\|\hat{f}_n - f_0\|_{L^2([0,1]^d)}^2\Big], \tag{56}$$

*where the infimum is over all estimators $\hat{f}_n$ based on $n$ i.i.d. observations from the regression model $Y = f_0(X) + \varepsilon$, $\mathbb{E}[\varepsilon \mid X] = 0$, $\mathbb{E}[\varepsilon^2] < \infty$. Then*

$$R_n(\mathcal{F}_r^{\text{KAN}}) \asymp n^{-\frac{2r}{2r+1}}, \tag{57}$$

*i.e., there exist constants $0 < c \leq C < \infty$ independent of $n$ such that*

$$c\, n^{-\frac{2r}{2r+1}} \;\leq\; R_n(\mathcal{F}_r^{\text{KAN}}) \;\leq\; C\, n^{-\frac{2r}{2r+1}}. \tag{58}$$

*In particular, the spline-based KAN sieve estimators of Theorems 1 and 2 are minimax optimal over $\mathcal{F}_r^{\text{KAN}}$.*

*Proof.* We first establish the upper bound and then the lower bound.

**Upper bound.** By Theorem 1, for any $f_0$ in the additive KAN subclass with $g_q, \psi_{qj} \in W^r([0,1])$, the spline-based KAN sieve estimator $\hat{f}_n$ satisfies

$$\mathbb{E}_{f_0}\big[\|\hat{f}_n - f_0\|_{L^2}^2\big] = O\big(n^{-\frac{2r}{2r+1}}\big), \tag{59}$$

when the number of spline knots is chosen as $k_n \asymp n^{1/(2r+1)}$. Similarly, Theorem 2 shows that the same estimator, applied to the hybrid (multiplicative) KAN architecture, attains the same rate under the same smoothness assumptions on $g_q$ and $\psi_{qj}$. Since $\mathcal{F}_r^{\text{KAN}}$ is contained in the union of these additive and hybrid subclasses, we obtain

$$\sup_{f_0 \in \mathcal{F}_r^{\text{KAN}}} \mathbb{E}_{f_0}\big[\|\hat{f}_n - f_0\|_{L^2}^2\big] \leq C\, n^{-\frac{2r}{2r+1}} \tag{60}$$

for some constant $C > 0$ independent of $n$. Taking the infimum over all estimators can only reduce the risk, hence

$$R_n(\mathcal{F}_r^{\text{KAN}}) \leq C\, n^{-\frac{2r}{2r+1}}. \tag{61}$$

**Lower bound.** To obtain a minimax lower bound for the entire class $\mathcal{F}_r^{\text{KAN}}$, it suffices to consider any smaller subclass contained within it. Let

$$W_r^{(1)} = \big\{ f : [0,1]^d \to \mathbb{R} \; : \; f(x) = h(x_1), \; h \in W^r([0,1]) \big\}, \tag{62}$$

i.e., functions depending only on the first coordinate via an $r$-smooth univariate function $h$. By the Kolmogorov–Arnold representation and the KAN construction, any such $f$ can be represented within our KAN architecture; for instance, take $Q = 1$, let the node be additive, set $g_1(u) = u$, choose $\psi_{11}(x_1) = h(x_1)$, and $\psi_{1j}(x_j) \equiv 0$ for $j > 1$. Hence

$$W_r^{(1)} \subset \mathcal{F}_r^{\text{KAN}}. \tag{63}$$

Consider the minimax risk over $W_r^{(1)}$,

$$R_n(W_r^{(1)}) = \inf_{\hat{f}_n} \sup_{f_0 \in W_r^{(1)}} \mathbb{E}_{f_0}\big[\|\hat{f}_n - f_0\|_{L^2}^2\big]. \tag{64}$$

Because the response depends only on $x_1$, this reduces to the classical univariate nonparametric regression problem with smoothness $r$. Standard minimax theory for Sobolev balls (see, e.g., Tsybakov, 2009) implies that there exists a constant $c > 0$ such that

$$R_n(W_r^{(1)}) \geq c\, n^{-\frac{2r}{2r+1}}. \tag{65}$$

Since $W_r^{(1)} \subset \mathcal{F}_r^{\text{KAN}}$, the minimax risk over $\mathcal{F}_r^{\text{KAN}}$ cannot be smaller than that over $W_r^{(1)}$, i.e.,

$$R_n(\mathcal{F}_r^{\text{KAN}}) \geq R_n(W_r^{(1)}) \geq c\, n^{-\frac{2r}{2r+1}}. \tag{66}$$

Combining equation 61 and equation 66 yields

$$c\, n^{-\frac{2r}{2r+1}} \;\leq\; R_n(\mathcal{F}_r^{\text{KAN}}) \;\leq\; C\, n^{-\frac{2r}{2r+1}}, \tag{67}$$

which proves the claimed minimax-optimal rate for KAN estimators over $\mathcal{F}_r^{\text{KAN}}$. $\qquad\square$

### A.4 Proof of Corollary 2

**Corollary 2** (Optimal knot number for spline-based KAN sieves). *Let $r > 1/2$, and suppose each univariate component $g_q$ and $\psi_{qj}$ in the KAN architecture is represented by a B-spline basis of order $m \geq r$ with $k_n$ interior knots, as in the sieve spaces $\mathcal{F}_n$ used in Theorems 1 and 2. Then the choice*

$$k_n \;\asymp\; n^{1/(2r+1)} \tag{68}$$

*balances the approximation error and estimation error in the KAN sieve estimator and yields the minimax-optimal convergence rate*

$$\mathbb{E}\big[\|\hat{f}_n - f_0\|_{L^2([0,1]^d)}^2\big] = O\Big(n^{-\frac{2r}{2r+1}}\Big). \tag{69}$$

*In particular, for each univariate spline unit approximating a component in $W^r([0,1])$, the resulting squared $L^2$ error satisfies*

$$\mathbb{E}\big[\|\hat{\psi}_{qj} - \psi_{qj}\|_{L^2([0,1])}^2\big] = O\Big(n^{-\frac{2r}{2r+1}}\Big) \tag{70}$$

*under the global knot choice equation 68.*

*Proof.* From the proofs of Theorems 1 and 2, the $L^2$ risk of the spline-based KAN sieve estimator admits the decomposition

$$\mathbb{E}\big[\|\hat{f}_n - f_0\|_{L^2}^2\big] \;\lesssim\; \underbrace{k_n^{-2r}}_{\text{approximation error}} + \underbrace{\frac{k_n}{n}}_{\text{estimation error}}, \tag{71}$$

where $k_n$ is the (common) number of interior knots used in the B-spline parameterization of each univariate component, and we have used that the effective sieve dimension satisfies $p_n \asymp k_n$.

The first term in equation 71 comes from spline approximation theory for univariate Sobolev functions of order $r$, applied to each $g_q$ and $\psi_{qj}$, and then propagated through the KAN composition by Lipschitz continuity on bounded domains. The second term comes from standard sieve least-squares theory, which yields $\mathbb{E}\|\hat{f}_n - f_n^*\|_{L^2}^2 = O(p_n/n) = O(k_n/n)$ for the empirical risk minimizer over the $p_n$-dimensional sieve space.

To obtain the optimal rate, we balance the two contributions by choosing $k_n$ to minimize $k_n^{-2r} + k_n/n$. This is achieved when

$$k_n^{-2r} \asymp \frac{k_n}{n} \quad\Longrightarrow\quad k_n^{2r+1} \asymp n \quad\Longrightarrow\quad k_n \asymp n^{1/(2r+1)}. \tag{72}$$

Substituting this choice into equation 71 gives

$$\mathbb{E}\big[\|\hat{f}_n - f_0\|_{L^2}^2\big] \;\lesssim\; n^{-\frac{2r}{2r+1}} + n^{-\frac{2r}{2r+1}} = O\Big(n^{-\frac{2r}{2r+1}}\Big), \tag{73}$$

which coincides with the minimax-optimal rate established in Corollary 1.

Finally, since the overall approximation error of $f_0$ is obtained by aggregating the spline approximations of the univariate components $g_q$ and $\psi_{qj}$, each such component inherits the same order of squared $L^2$ error under the common knot choice $k_n \asymp n^{1/(2r+1)}$, up to constants depending only on $Q$ and $d$. This yields the stated bound for $\mathbb{E}\|\hat{\psi}_{qj} - \psi_{qj}\|_{L^2}^2$ and completes the proof. $\qquad\square$

