# OpenReview forum: "On the Rate of Convergence of Kolmogorov-Arnold Network Regression Estimators"
_ICLR.cc/2026/Conference — Submitted to ICLR 2026_

### Official Review · Reviewer_Gn47 · 2025-10-17

**Soundness:** 3
**Presentation:** 3
**Contribution:** 3
**Rating:** 6
**Confidence:** 4

**Summary:**

This paper presents a theoretical framework for analyzing the convergence rate of Kolmogorov-Arnold Networks on Sobolev spaces of smoothness r. The scaling exponent $2r/(2r+1)$ highlights that KANs do not suffer from curse of dimensionality because the scaling exponent for MLPs is $2r/(2r+d)$ where $d$ is the input dimensionality. The authors further derive how to optimally choose grid size k and support their conclusions with numerical experiments. These results provide a theoretical foundation for using KANs in non-parametric regression.

**Strengths:**

* The paper is well motivated and nicely written
* The paper presents a theoretical contribution on the convergence rate of KANs
* Theoretical claims are supported by numerical experiments

**Weaknesses:**

* Generalization of the result to deeper KANs. Theorem 1 and 2 seemed to assume two-layer KANs. Can this result be extended to deeper KANs?
* The optimal number of knots $k$ is not supported by numerical experiments.

**Questions:**

* Theorems 1 and 2 seemed to assume two-layer KANs. Can this result be extended to deeper KANs?
* In Corollary 2, the optimal number of internal knots is derived. For smooth functions (say $f(x) = {\rm sin}(x)$) when $r\to\infty$, how should one choose the optimal $k$?
* In abstrac, $n$ is used without defnition.
* a nitpick, since $k$ is reserved for the optimal knot number, it's better to use a different subscript letter in $a_k$ (line 402).

---

> ### Author Response · Authors · 2025-11-24
>
> We sincerely thank the reviewer for the careful reading and constructive feedback.
>
> # Weakness 1 & Question 1
> The analysis focuses on the canonical Kolmogorov–Arnold representation form. Extending the results to deeper architectures is possible in principle, but would require additional assumptions on how smoothness propagates through multiple nested compositions. To keep the theory focused and sharp, we restrict our analysis to the foundational KAN model.
>
> # Weakness 2
> In the current submission, the purpose of the numerical study is to verify the theoretical convergence exponent rather than to evaluate model selection strategies. Since the focus of the paper is on establishing the minimax rate and deriving the theoretical guideline, the experiments fix the number of knots according to this theoretical rule for each sample size. This ensures that the empirical results directly reflect the theorem, without introducing variability from additional model-selection procedures.
>
> # Question 2
> Corollary 2 derives the optimal knot number  $k_n \asymp n^{1/(2r+1)}$ under the assumption that the target function has *finite* Sobolev smoothness $r$. Thus, the result is intended to be interpreted for any fixed $r < \infty$. In this regime, $k_n$ increases with $n$ at the appropriate rate to balance approximation and estimation errors.
>
> The expression $k_n \asymp n^{1/(2r+1)} \to 1$ as $r \to \infty$ should not be taken literally as recommending one knot. Rather, this limiting behavior reflects the intuitive fact that as a function becomes extremely smooth, fewer knots are needed for optimal spline approximation. In the limiting case $r \to \infty$, the optimal spline space degenerates to a single global polynomial (possibly of high order), which corresponds to the classical idea of approximating an infinitely smooth function via its Taylor expansion.
>
> # Qeustion 3 & 4
> Thank you for the careful reading. These notation issues have been corrected in the revised manuscript.

---

### Official Review · Reviewer_UAqq · 2025-10-27

**Soundness:** 2
**Presentation:** 3
**Contribution:** 2
**Rating:** 2
**Confidence:** 4

**Summary:**

This paper considers estimating functions from a certain model class using KANs. The model class is closely related to the KAN architecture and essentially consists of functions which can be expressed using KANs with a certain amount of smoothness. The resulting rates are shown to be minimax optimal, and numerical experiments are performed. I think that the paper has quite an interesting potential idea, however, it seems to me that the authors have not described the estimator that they are analyzing. This issue would need to be addressed for the results to make sense.

**Strengths:**

Function estimation using KANs is becoming increasingly popular, so the problem addressed is of significant importance. The paper is also well-presented and written.

**Weaknesses:**

There are appear to be some significant technical details missing. Specifically, in Theorem 1 what exactly is the estimator $\hat{f}$? You say that $\hat{g}\_q$ and $\hat{\psi}\_{qj}$ are constructed via least squares with back-fitting. What does this mean, exactly? I can't find an explanation elsewhere in the paper. Note that only the composite function $f$ is sampled, you do not have access to the true $g_q$ and $\psi_{qj}$ to do regression on. This point confused me a bit, could the authors please be explicit about what estimator they are studying.

**Questions:**

Could the authors also please clarify what algorithms are used for training their KANs in the experimental results section? If these algorithms differ from the estimator which is theoretically analyzed, please indicate this.

If these issues about the precise definition of the estimator are addressed, I would be willing to raise my score.

---

> ### Author Response · Authors · 2025-11-23
>
> We sincerely thank the reviewer for the thoughtful feedback. The reviewer correctly noted that the original submission described the optimization procedure using backfitting. We would like to clarify that backfitting was an optional numerical solution for approximating the internal univariate components. We now recognize that its presence in the description could unintentionally give the impression that the theoretical guarantees rely on access to the true inner functions or on separate regression steps for the univariate units.
>
> In the revised version, we have made the formulation of the estimator precise and fully aligned with the practical implementation. Both theory and experiments now treat KAN training as the **empirical risk minimization over the spline-parameterized KAN function class**. The estimator under analysis is now defined explicitly as the **spline-based KAN sieve estimator**, and the convergence proofs have been fully updated based on classical sieve theory, rather than backfitting or sequential regression of inner functions.
>
> This makes it explicit that the optimization updates all internal functions simultaneously, and no regression access to the true inner components is assumed anywhere in the theoretical development. As a result, the current theory and implementation are unified. The model analyzed in the main results is the model trained in the experiments.
>
> Because the theoretical estimator is now fully expressed as a **standard ERM over the KAN sieve space**, references to backfitting no longer describe the implemented method and have been removed to prevent misunderstanding. The Section 5 Simulation Study has also been updated to clarify that the LBFGS optimizer numerically minimizes this very same least-squares objective, so that the estimator studied in theory matches the estimator trained in practice.
>
> We are grateful to the reviewer for pointing us toward this clarification, and Section 3 Model Setup has been updated with a paragraph that makes this explicit.

---

> > ### Comment · Reviewer_UAqq · 2025-11-25
> >
> > I thank the authors for the clarification. I just have one more question, which is the following. In the updated paper, you say that the spline-parameterized KAN functions form a finite dimensional linear space. It seems that this cannot be true unless you only train the outer functions $g_q$ while leaving the inner function $\psi_{qj}$ fixed. Is this what you are doing? It appears to me that in the rest of the paper you train both the inner and outer functions. In this case, I don't think that this is correct.

---

> > > ### Author Response · Authors · 2025-11-26
> > >
> > > Thank you for the insightful follow-up question. We would like to clarify that both inner and outer functions are trained jointly, and that the paper did not claim that the spline-parameterized KAN function class is a linear space, as stated around line 251:
> > > > For a given sample size $n$, the collection of all such spline-parameterized KANs spans a finite-dimensional function class.
> > >
> > > The property we rely on in the theoretical development is that the hypothesis class is finite dimensional for each sample size $n$. When parameterizing $g_q$ and $\phi_{qj}$ with a B-spline basis of $k_n$ knots, the overall KAN model is parameterized by a finite number of spline coefficients. In this sense, for each sample size $n$, the hypothesis class forms a finite-dimensional parametric family. It is not linear because compositions and products break linearity.
> > >
> > > This distinction matters because sieve least-squares theory requires a finite-dimensional growing hypothesis space. It does not require, which we also did not claim, linearity.
> > >
> > > We appreciate the reviewer for highlighting this subtle point, and the opportunity to clarify it. If there is anything that would benefit from further elaboration, we would be very glad to provide it.

---

### Official Review · Reviewer_rrB8 · 2025-10-29

**Soundness:** 3
**Presentation:** 3
**Contribution:** 2
**Rating:** 4
**Confidence:** 4

**Summary:**

The paper studies the statistical behavior of Kolmogorov–Arnold Networks (KANs) in the nonparametric regression. KANs are based on the Kolmogorov–Arnold representation theorem, which guarantees that any continuous multivariate function can be expressed as a sum of univariate functions applied to linear combinations of input variables.

The paper focuses on a setting of KANs where each univariate component is modeled using B-spline basis functions. The paper then places this architecture within the framework of classical nonparametric estimation and provides a complete analysis of its convergence properties under standard smoothness assumptions.

The main result shows that, under Sobolev smoothness of order r, both additive and hybrid (additive–multiplicative) KAN models achieve the same minimax-optimal mean-squared convergence rate as traditional spline or kernel estimators. The paper also offers a practical rule for choosing the number of spline knots and validates the theoretical predictions with some numerical experiments.

**Strengths:**

The major contribution of this paper is that for both additive and hybrid KAN estimators, they obtained the minimax-optimal mean-squared error rate under Sobolev smoothness assumptions. This places the models on a solid theoretical foundation within nonparametric statistics.

Moreover, using these results, the paper provides the guidance on the choice of the optimal number of spline knots $k \asymp n^{1/(2r+1)}$.

**Weaknesses:**

The main theorems establish the classical minimax rate for Sobolev regression functions. While they considered Kolmogorov–Arnold Networks (KANs) in the nonparametric regression, they only considered additive and hybrid (additive–multiplicative) KAN models, see Equation (11) in the paper. Using B-spline basis functions to obtain minimax optimal regression estimators for Sobolev regression functions is standard and has been studied in many existing papers. Due to this simple form of the model in Equation (11), it is not clear if this is a significant contribution to the statistical and machine learning community at ICLR.

Moreover, the paper [1] studied minimax optimality of a more complex model, the deep composition of anisotropic Besov functions, which seems to cover the setting of this paper, at least, for the theoretical contributions and using B-splines. I think the theoretical contributions of this paper then may not be adequate. Also, in recent regression literature such as [1,2,3], adaptivity to unknown smoothness parameters are usually studied. While this paper provided a suggestion of picking the optimal number of spline knots, it relies on the unknown parameter r.

Overall, I think this paper appears to be a subset of [1], or has a lot of overlaps with [1], making its theoretical contributions unclear. Its contributions seem standard and combinatorial.

[1] Suzuki, Taiji, and Atsushi Nitanda. "Deep learning is adaptive to intrinsic dimensionality of model smoothness in anisotropic Besov space." Advances in Neural Information Processing Systems 34 (2021): 3609-3621.

[2] Shi, Zhaoyang, Krishna Balasubramanian, and Wolfgang Polonik. "Adaptive and non-adaptive minimax rates for weighted Laplacian-Eigenmap based nonparametric regression." International Conference on Artificial Intelligence and Statistics. PMLR, 2024.

[3] Chazal, Frédéric, Ilaria Giulini, and Bertrand Michel. "Data driven estimation of Laplace-Beltrami operator." Advances in Neural Information Processing Systems 29 (2016).

**Questions:**

I may have some misunderstand of the paper but please clarify the questions from Weaknesses.

**Details Of Ethics Concerns:**

None.

---

> ### Author Response · Authors · 2025-11-23
>
> We thank the reviewer for the careful reading and constructive criticism.
>
> Regarding the concern that the model in Equation (11) may look too simple, the simplicity is an explicit feature of the Kolmogorov–Arnold representation. KANs are defined precisely through sums of univariate nonlinear transformations applied to structured combinations of inputs, with both additive and multiplicative aggregation permitted at each node. This structure is directly aligned with the Kolmogorov–Arnold representation theorem rather than a generic additive model.
>
> Regarding the paper [1], the two papers study fundamentally different function classes. The paper [1] analyzes deep ReLU neural networks operating over anisotropic Besov spaces, where the smoothness varies by direction and the target functions are multivariate and deeply compositional. Our paper investigates functions representable by the Kolmogorov–Arnold architecture: finite sums of Sobolev-smooth univariate components composed through additive/multiplicative aggregations. Importantly, the multivariate deep-composition model in [1] cannot express the KAN structure unless each multivariate layer degenerates into a set of completely separable univariate functions. Even under this hypothetical degeneration, the model in [1] would yield independent coordinate-wise pipelines without cross-coordinate aggregation, and therefore still would not recover the additive/multiplicative interactions that define KANs. Likewise, KANs include functions that follow the Kolmogorov–Arnold decomposition but do not satisfy the directional anisotropy and deep multivariate smoothness assumptions needed in [1]. Therefore, the two function classes are structurally non-overlapping, and neither subsumes the other. We will make this distinction explicit in the revised related-work section to avoid the impression that the theoretical contributions follow directly from [1].
>
> In addition to the theoretical analysis, the paper includes simulation studies that verify the rate numerically. The experiments track the decay of mean-squared error against sample size on controlled function classes of known Sobolev smoothness. Both additive and hybrid KANs follow the predicted slopes, supporting the theoretical convergence guarantees.

---

### Official Review · Reviewer_TYLA · 2025-10-31

**Soundness:** 3
**Presentation:** 3
**Contribution:** 3
**Rating:** 6
**Confidence:** 2

**Summary:**

This paper establishes the theoretical convergence rate of Kolmogorov–Arnold Networks (KANs) in nonparametric regression. By representing the univariate components of KANs with B-splines, the authors connect KANs to classical sieve estimation theory. They prove that both additive and hybrid KANs achieve the minimax-optimal convergence rate under Sobolev smoothness assumptions. Experiments on synthetic data confirm that the empirical convergence behavior matches the theoretical rate.

**Strengths:**

1. This paper provides a complete theoretical treatment of KAN convergence rates, bridging the empirical success of KANs with rigorous nonparametric learning theory. This significantly advances the theoretical foundation of KAN research.

2. By parameterizing univariate KAN components using B-splines, the authors successfully unify KANs with the sieve estimation framework, showing that structured neural architectures can inherit the statistical optimality of spline estimators while maintaining interpretability.

3. The simulation experiments are well aligned with the theoretical analysis, showing empirical slopes near the theoretical exponent

**Weaknesses:**

1. The theoretical results rely on restrictive assumptions, such as all univariate functions belonging to Sobolev spaces with known smoothness and bounded constants, which may limit their applicability to practical, high-dimensional learning settings.

2. The experimental validation focuses solely on smooth synthetic functions under low-noise settings. No results are provided for higher-dimensional, noisy, or real-world datasets, limiting the empirical generality of the conclusions.

3. The paper compares KANs only to standard MLPs. Including baselines such as DeepONet, SIREN, or kernel regression networks would more clearly demonstrate the practical advantages of KANs in terms of interpretability, stability, and efficiency.

**Questions:**

See the weakness.

---

> ### Author Response · Authors · 2025-11-23
>
> We thank the reviewer for the thoughtful and encouraging assessment of our work and for the helpful suggestions to further strengthen the paper.
>
> # Weakness 1
> The current analysis adopts Sobolev smoothness with known regularity primarily to establish a rigorous statistical foundation for KAN estimators. This assumption follows standard practice in nonparametric convergence theory for spline-based sieves. Our intention is not to imply that all practical applications operate under these exact conditions, but to provide a theoretically sound baseline on which more flexible variants can be developed.
>
> # Weakness 2
> The synthetic experiments in the paper were designed deliberately to isolate smoothness-controlled regression functions so that the decay of error with respect to sample size could be measured precisely and compared to the theoretical exponent. This controlled setup is standard in convergence-rate validation and allows a direct empirical test of the theoretical result. We appreciate the suggestion to explore additional scenarios, and we have now clarified in the text that the goal of the experiments is to verify the established rates, rather than to provide an exhaustive empirical benchmark suite.
>
> # Weakness 3
> In the introduction, we discuss known theoretical rates for fully connected neural networks, which are typically slower (up to logarithmic factors) than the minimax rate achieved by spline-based estimators. Recent empirical studies on KANs have also reported that KANs often converge faster and require fewer samples than standard MLPs. Our comparison with MLP is therefore designed to connect these observations with our theory by showing that a structured KAN estimator attains the optimal rate $O(n^{-2r/(2r+1)})$ in exactly the setting where classical fully connected networks are known to be slower in theory and often in practice. We have now made this rationale explicit in the paper and clarify that the goal of the empirical section is to illustrate this theoretical contrast, while a comprehensive empirical comparison with other architectures is left for future work focused on applications.
>
> In response to Weaknesses 2 and 3, we have added a clarifying paragraph at the beginning of Section 5:
> >To verify the theoretical results and demonstrate the practical performance of Kolmogorov--Arnold Networks (KANs), we conduct simulation experiments on synthetic datasets generated from functions with known smoothness. It is noted that this paper focuses on theoretical convergence rather than empirical benchmarking, and the experiments are intentionally designed to validate that the empirical convergence rate matches the theoretical exponent. In particular, we compare additive KANs and hybrid additive--multiplicative KANs against standard multilayer perceptrons (MLPs). Including the MLP baseline serves to verify the theoretical contrast between KAN and conventional neural networks in terms of convergence behavior.

---

### Author Response · Authors · 2025-12-02
**Summary of Rebuttal**

We thank all reviewers for their careful reading and constructive feedback, and we are grateful to the area chair for taking over the evaluation of our submission under the exceptional circumstances of this review cycle. Below is a brief, factual summary of the reviewers' concerns and how the rebuttals have addressed them.

# Reviewer UAqq (estimator definition)
The initial review raised a concern about the definition of the estimator analyzed in Theorem 1, and the reviewer **explicitly indicated willingness to raise the score** once this point was clarified. In the revised version, we have added a paragraph in Section 3 (Model Setup) to further clarify the definition and have made the estimator fully explicit as the **spline-based KAN sieve estimator**, i.e., the empirical risk minimizer over the finite-dimensional class of spline-parameterized KANs. We have also clarified that both inner and outer functions are trained jointly, and that no access to true inner functions is assumed.

Following our clarification, the reviewer **raised one additional short question** regarding whether we were implicitly assuming that the hypothesis class of spline-parameterized KANs is linear. We clarified that we do not assume linearity, nor is linearity required for the proof. The argument relies only on the fact that the hypothesis class is finite-dimensional for each sample size, which is the standard requirement for sieve least-squares theory.

# Reviewer rrB8 (relation to existing literature)
This reviewer questioned whether our theoretical results are essentially a subset of Suzuki & Nitanda (NeurIPS 2021). In our rebuttal, we clarified that the **two works study different function classes and architectures**. Suzuki & Nitanda analyzed deep ReLU networks on anisotropic Besov spaces with multivariate compositional structure, whereas we analyze Kolmogorov–Arnold Networks with finite sums of univariate Sobolev-smooth components combined through additive and multiplicative aggregation. **Neither function class contains the other**, so our convergence results and the derived optimal knot scaling do not follow from Suzuki & Nitanda (2021). This conceptual distinction has been made explicit in the rebuttal.

# Reviewers Gn47 and TYLA (scope and empirical study)
Both Reviewers Gn47 and TYLA **evaluated the work positively and raised questions mainly about scope and empirical coverage**.

* For Reviewer Gn47, we explained that extending the results to deeper KANs is conceptually possible but would require additional assumptions on smoothness propagation through multiple compositions, and is therefore left outside the current scope to keep the theory focused on the canonical form as prescribed by the Kolmogorov–Arnold representation. We also clarified that the experiments fix the number of knots according to the theoretically derived rule to isolate the rate behavior, and we corrected the minor notation issues the reviewer pointed out.

* For Reviewer TYLA, we clarified that the Sobolev assumptions are standard in nonparametric convergence theory, and that the synthetic experiments are intentionally designed to verify the convergence exponent in a controlled setting, not to serve as a broad empirical benchmark against many architectures. The MLP baseline was chosen specifically to highlight the theoretical contrast in convergence rates.

We hope this brief summary helps clarify how the existing rebuttals address the key points raised in the reviews, especially in light of the recent freeze on further reviewer discussion. We would, of course, be very gald to provide any additional technical details if needed.

---

### Meta-Review · Area_Chair_o3cs · 2026-01-07

**Summary:**

This paper presents a theoretical analysis of Kolmogorov–Arnold Networks (KANs) with univariate spline units for nonlinear regression. The authors show that both additive and hybrid additive–multiplicative KAN architectures with the sieve estimator achieve minimax-optimal convergence rates under Sobolev smoothness assumptions.

Reviewers generally appreciate the presented theoretical convergence rate. However, I share the concerns about (i) Limited novelty: the theoretical analysis uses standard results from spline approximation theory and empirical-process arguments for sieve least square and it is only carried on 2-layer KAN; (ii) missing comparison with other baselines.

**Reviewer Concerns:**

Some of the concerns by reviewers include

- Using B-spline basis functions to obtain minimax optimal regression estimators for Sobolev regression functions is standard and has been studied in many existing papers: Reviewer rrB8. In fact, this is quite apparent in the proofs of the main results.

- The current results are only for 2-layer KANs: Reviewer Gn47. In their rebuttal, the authors respond that the analysis can be extended to deeper KANs, but it is not clear if/how it could be done.

- The paper compares KANs only to standard MLPs, missing baselines such as DeepONet, SIREN, or kernel regression:Reviewer TYLA.
The authors respond that their purpose is to verify theoretically the empirical performance of KAN compared with MLP.

- Reviewer UAqq asked for clarification regarding the estimator $\hat{f}$, which has been clarified by the authors in their response.

**Reviewer Scores:**

The scores are 6,4,2,6

Reviewer UAqq (2) would have raised their score since the authors have clarified the definition of the estimator. The other reviewers would not have changed their scores much.

---

### Decision · Program_Chairs · 2026-01-26

Reject